# Impact of Alcohol Consumption on Male Fertility Potential: A Narrative Review

**DOI:** 10.3390/ijerph19010328

**Published:** 2021-12-29

**Authors:** Renata Finelli, Filomena Mottola, Ashok Agarwal

**Affiliations:** 1American Center for Reproductive Medicine, Cleveland Clinic, Cleveland, OH 44106, USA; finelli.renata@gmail.com; 2Department of Environmental, Biological and Pharmaceutical Sciences and Technologies, University of Campania Luigi Vanvitelli, 81100 Caserta, Italy; filomena.mottola@unicampania.it

**Keywords:** alcohol-related disorders, ethanol, ethyl alcohol, infertility, male, spermatozoa

## Abstract

Alcohol abuse disorder is a serious condition, implicating more than 15 million people aged 12 years and older in 2019 in the United States. Ethanol (or ethyl alcohol) is mainly oxidized in the liver, resulting in the synthesis of acetaldehyde and acetate, which are toxic and carcinogenic metabolites, as well as in the generation of a reductive cellular environment. Moreover, ethanol can interact with lipids, generating fatty acid ethyl esters and phosphatidylethanol, which interfere with physiological cellular pathways. This narrative review summarizes the impact of excessive alcohol consumption on male fertility by describing its metabolism and how ethanol consumption may induce cellular damage. Furthermore, the impact of alcohol consumption on hormonal regulation, semen quality, and genetic and epigenetic regulations is discussed based on evidence from animal and human studies, focusing on the consequences on the offspring. Finally, the limitations of the current evidence are discussed. Our review highlights the association between chronic alcohol consumption and poor semen quality, mainly due to the development of oxidative stress, as well as its genotoxic impact on hormonal regulation and DNA integrity, affecting the offspring’s health. New landscapes of investigation are proposed for the identification of molecular markers for alcohol-associated infertility, with a focus on advanced OMICS-based approaches applied to the analysis of semen samples.

## 1. Introduction

The consumption of alcoholic beverages is common in numerous societies, and almost 60% of the global population aged 15 years and over have been reported consuming alcoholic drinks in one year [1]. In Europe, 76% of citizens had reportedly consumed alcoholic beverages in the one year preceding a survey conducted by the European Commission [2]. Binge drinking is characterized by drinking several alcoholic beverages in a short period of time, where the blood alcohol concentration is around 0.08 g/dL [3]. According to the National Institute on Alcohol Abuse and Alcoholism, around 26% of the adult population reported binge drinking in the United States, with almost 15 million people older than 12 years showing alcohol abuse disorder [4]. This behavior is estimated to affect 4% of the adult global population [5].

Heavy alcohol consumption (defined as more than 3 and 4 drinks in a day, or more than 7 and 14 drinks weekly, for women and men, respectively) is reported to negatively affect human health, promote traffic accidents, and alter social behaviors, with severe repercussions for personal, social, and professional lives [5]. Clinically, alcohol consumption has been correlated with an increased incidence of different types of cancer [6], cardiovascular [7] and liver diseases [8], birth defects [9], and psychiatric disorders [10].

Alcohol abuse may result in alcohol dependence: this is a complex and dynamic process involving biological and socioenvironmental factors [11,12,13]. In general, alcohol acts by suppressing the activity of the nervous system. This is evident upon withdrawal after a prolonged period of abuse, where abstinence is characterized by the hyperactivation of the system itself, with consequent tachycardia, hypertension, excessive sweating, tremors, and convulsions [14]. Other common manifestations include rapid mood changes, irritability, agitation and anxiety, sleep disturbances, and the inability to experience pleasure (anhedonia), as well as a reduction in the pain threshold [15,16]. 

Numerous clinical studies have been conducted on alcoholic subjects to define the effects of alcohol abuse on the male reproductive capacity [17,18,19,20]. Initially, there can be alterations of the secondary sexual characteristics, such as reduction in facial and body hair, enlargement of the breasts, and accumulation of adipose tissue from the abdominal area to the hips [17]. This is followed by the establishment of a proinflammatory status, indicated by elevated levels of leukocytes in the semen [18]. From a functional point of view, alcoholics often complain of sexual dysfunction or infertility [19,20]. As a large percentage of male alcoholics are of the reproductive age and may be interested in becoming fathers [1], understanding the mechanisms leading to alcohol-related male infertility is of the utmost importance.

This narrative review aims to discuss the impact of alcohol consumption on male fertility by describing alcohol metabolism and the molecular mechanisms by which ethanol (EtOH) can induce cellular damage. Moreover, we discuss evidence from animal and human studies regarding the impact of EtOH consumption on sexual hormonal regulation, semen quality, and epigenetics, emphasizing the consequences on the offspring. Finally, we highlight the limitations of the current evidence and describe new areas of investigation for the identification of the molecular markers of alcohol-associated male infertility.

## 2. Metabolism of Ethanol

When an alcoholic drink is consumed, the EtOH is rapidly absorbed through the stomach (20%) and duodenum (80%), and passes directly into the bloodstream [21]. Through the blood circulation, alcohol reaches the liver, brain, heart, and kidneys within 10–15 min after consumption, while it takes about an hour to reach the muscles and adipose tissue, where it tends to concentrate [22].

The liver metabolizes about 90% of the EtOH, while the remaining EtOH is directly eliminated through urine, feces, breast milk, sweat, and exhaled air [21]. In the liver, stomach, and intestinal tract, EtOH is oxidized primarily by alcohol dehydrogenase (ADH) into acetaldehyde, a highly toxic and reactive molecule, with a reduction in the cofactor, nicotinamide adenine dinucleotide (NAD^+^) [23] (Figure 1). The reduced cofactor (NADH) contributes to the cellular energy generation by means of the electronic transport chain located in the inner membrane of the mitochondria; through this pathway, the metabolism of EtOH produces energy (7 kcal/g of EtOH) [24]. 

In the brain, the enzymes, catalase and cytochrome P450 (CYP450), oxidize 65% and 20% of the EtOH into acetaldehyde (Figure 1), respectively [25]. CYP450 is a family of enzymes containing the heme group that are involved in the oxidation of steroids, fatty acids, and numerous xenobiotics of environmental origin. In particular, the cytochrome P450 2E1 (CYP2E1) activity is induced by chronic alcohol consumption, resulting in increased alcohol tolerance, oxidative stress and toxicity, with a consequently higher risk of developing cancer and liver injury [26]. Similarly, catalase, which belongs to the class of oxidoreductases, contains a heme group and is generally involved in the cellular scavenging of reactive oxygen species (ROS) [21]. The enzyme is localized in the peroxisomes, and it oxidizes EtOH in the presence of complexes that generate H_2_O_2_, such as reduced nicotinamide adenine dinucleotide phosphate (NADPH) oxidase or xanthine oxidase (Figure 1). Whereas catalase plays a marginal role in EtOH oxidation in the liver, it is involved in the development of alcohol tolerance and consumption in the brain [27].

The enzyme, acetaldehyde dehydrogenase (ALDH), oxidizes acetaldehyde and produces acetate. ALDH is a zinc-dependent enzyme, which also uses NAD^+^ as a cofactor [21] (Figure 1). Some drugs used to treat alcoholism, such as Disulfiram, inhibit the activity of ALDH, resulting in the accumulation of acetaldehyde. This leads to unpleasant effects, such as increased sweating and tachycardia, nausea, and vomiting, thus discouraging the consumption of alcohol [28]. Acetate, which results from the oxidation of acetaldehyde, leads to increased blood flow in the liver and a depressed central nervous system, and it compromises numerous metabolic processes [29,30]. Acetate can be metabolized into acetyl-CoA, a key molecule in the synthesis of lipids and cholesterol in the mitochondria. Depending on the nutritional, energetic, and hormonal status, acetyl-CoA can be converted into CO_2_, fatty acids, ketones, or cholesterol [21]. 

The nonoxidative metabolism of alcohol is minimal, but its products can have pathological and diagnostic significance. In fact, EtOH can react with fatty acids, forming fatty acid ethyl esters [31]. These esters are synthesized in the endoplasmic reticulum and are transported to the membranes, where they bind to lipoproteins or albumin. They are potentially toxic, as they interfere with the synthesis of DNA and proteins [21]. Furthermore, EtOH can be incorporated into phospholipids as phosphatidylethanol through a reaction catalyzed by the enzyme, phospholipase D (PLD) [32]. PLD has a high Michaelis constant (Km) for ethanol, so that the enzymatic reaction occurs mainly at high concentrations of circulating alcohols. Phosphatidylethanol is poorly metabolized, and can accumulate after chronic alcohol intake, interfering with the cellular signaling pathways [32].

## 3. Ethanol-Induced Mechanisms of Cellular Damage

EtOH consumption has been reported to induce cellular damage through different interconnected mechanisms, which include the establishment of an inflammatory and oxidative environment, resulting in genotoxicity and enhanced apoptotic rate (Figure 2). 

The close link between chronic alcohol consumption and the onset of inflammation has been widely investigated [33,34,35,36,37,38,39,40,41,42]. In the liver, acetaldehyde activates inflammatory signaling pathways by increasing the synthesis of circulating proinflammatory cytokines by Kupffer cells [33], such as tumor necrosis factor-alpha (TNFα) [33]. TNFα acts on the hepatic stellate cells, further exacerbating the synthesis of proinflammatory and profibrogenic molecules, impairing cell function, and contributing to alcohol-induced cirrhosis [33,34,35]. In the liver, brain, and intestine, chronic alcohol administration has been proven to activate the inflammasomes. This is an intracytoplasmic multiprotein complex that induces the activation of caspase-1 and interleukin (IL)-1β, enhancing the proinflammatory status in the same tissues [36,37]. Moreover, through a mechanism of negative feedback, inflammation enhances the negative impact of alcohol metabolism through the reduced expression of ADH1 and ALDH2, which causes systemic and hepatic increases in the EtOH and acetaldehyde levels [40]. In addition, the metabolism of acetaldehyde into acetate increases the NADH:NAD^+^ ratio within the cell, and consequently inhibits the fatty acid β-oxidation pathway in the liver. This results in the accumulation of triglycerides in the hepatocytes as hepatic steatosis, and the enhancement of inflammation [38]. 

Considering that inflammatory pathways participate in the pathophysiology of neurodegenerative disorders, an alteration of the cytokine expression in the nervous system can severely damage neuron activity [41]. EtOH exposure alters the NF-kB pathway, which is important in inflammatory and immune responses in the brain, and decreases the transcription of regulatory cyclic protein AMP-responsive element-binding protein (CREB), affecting neuronal survival and protection from apoptosis [42]. 

Moreover, alcohol consumption can increase oxidative stress through different mechanisms, with serious consequences for human health. ROS play a crucial role as mediators in several biological processes, including cell signaling and pathogen resistance [43]. However, ROS are highly unstable and are reactive with proteins, lipids, and any cellular structures. When ROS are overproduced, they switch the cellular redox status towards oxidative stress, which contributes to several pathological conditions [44].

EtOH metabolism enhances ROS generation, as the activity of dehydrogenases and the CYP2E1 system increases the levels of NADH, which, in turn, is oxidized by xanthine oxidase, with the generation of ROS [45]. An increased oxidative status is also due to the increased peroxisomal activity in the livers of heavy drinkers, as a result of steatosis [46]. 

In addition to directly increasing the production of ROS, in vitro studies have shown that acetaldehyde reportedly causes an intracellular redox imbalance by increasing the levels of oxidative stress markers (i.e., malondialdehyde (MDA)), and by reducing the concentration of antioxidant glutathione (GSH) [47,48]. Acetaldehyde also impairs the enzymatic activity of superoxide dismutase (SOD) 2, a powerful endogenous antioxidant [49]. In alveolar macrophages, chronic alcohol consumption and the subsequent onset of oxidative stress alter the expression of the NADPH oxidases (Noxes), enzymes contributing to the phagocyte-mediated host defense [50]. EtOH consumption upregulates the expression of the Noxes, as well as the regulatory proteins, p22phox, p47phox, and p67phox, both in vivo and in vitro, resulting in the higher intracellular production of O₂_−_ and H_2_O_2_, and lower GSH levels in the lung tissue [50].

The establishment of an inflammatory and oxidative environment modifies cellular homeostasis and DNA integrity, and can have a genotoxic/mutagenic impact, eventually leading to cell death. In vitro and in vivo studies indicate that EtOH induces autophagy in neurons by inhibiting the expression of the antiapoptotic Bcl-2 family proteins, while it increases the expression of proapoptotic proteins [51]. The expression of Bcl-2 is also modulated by EtOH through epigenetic regulation. In fact, by increasing the histone deacetylase activity, EtOH affects the binding of acetyl-histone H3 to the Bcl-2 promoter, which results in a reduced Bcl-2 gene transcription [52]. Furthermore, it has been suggested that EtOH induces programmed cell death through the mitochondrial pathway, as the inhibition of caspase-9 (key molecule in this pathway) resulted in a reduced apoptotic rate [53]. The activity of caspase-3, an apoptotic marker, was also increased in the cerebral cortices of rats after EtOH treatment [54], possibly because of the depolarization of the mitochondrial membrane by acetaldehyde accumulation [55]. The role of alcohol metabolites as mediators of mitochondrial damage was also demonstrated in knock-out mice for ALDH or ADH enzymes, which showed reduced membrane potential in comparison to the controls [55]. 

However, whereas evidence shows a negative impact of high-dose EtOH on cellular physiology, a low dose seems to reduce the apoptotic rate. In fact, recent data suggests that a low dose of EtOH contributes to the regulation of the expression of the TNF receptor, p75NTR, and a potassium-chloride transporter (KCC2) [56]. While p75NTR is involved in the regulation of apoptosis, KCC2 regulates the cellular electrochemical gradient, and its expression is crucial to maintaining the classic hyperpolarizing GABAergic inhibition in mature adult neurons [56].

EtOH and its metabolites, including ROS, can be classified as genotoxic agents. In fact, they enhance the rate of the DNA strand breaks, leading to chromosomal rearrangements and/or the loss of genetic information, genomic instability, and micronucleus formation [57]. Studies on human peripheral blood lymphocytes show mutagenic activity in alcoholics, with an increased sister chromatid exchange frequency, indicating alterations in the DNA replication or repair processes [58]. Acetaldehyde and other toxic aldehydes produced by lipid peroxidation, such as MDA and 4-hydroxynonenal (4-HNE), damage DNA, as is described in both animal and human studies [59,60]. These molecules can increase the generation of DNA adducts, such as those with the deoxyguanosine (dG) base (N2-ethyl-2′-deoxyguanosine, 1,N2-propane-2′-dG (PDG), and propane-dG), damaging the integrity of the DNA double helix [61]. Moreover, MDA and 4-HNE form etheno-DNA adducts, with strong carcinogenic mutagenic properties [60]. The most relevant mutagenic activity is ROS-mediated, as high levels of ROS induce the oxidation of DNA and the production of various lesions, including oxidized bases (8-hydroxy-2′-deoxyguanosine), apurinic/apyrimidine sites, and single or double DNA strand breaks [62,63]. ROS-related DNA damage is also associated with accelerated telomere shortening, since shortened telomeres were reported in chronic inflammatory diseases, compared to healthy individuals (as comprehensively described in [64]). Several studies have reported a cause–effect association between early telomere shortening and EtOH consumption [65,66]. However, the mechanistic link between EtOH metabolism and telomere shortening is unclear and requires further investigation.

## 4. Alcohol Consumption and Male Infertility: Evidence from Animal and Human Studies

Several animal and human studies have investigated the impact of alcohol consumption on reproductive hormonal regulation, semen quality, gene transcription, genetics, and epigenetics regulation, as well as the transgenerational impact of paternal exposure on the offspring. Evidence is discussed below and is summarized in Table 1.

### 4.1. Impact of Alcohol on Reproductive Hormonal Regulation

The secretion of reproductive hormones, such as testosterone, is regulated by the hypothalamic–pituitary–gonadal (HPG) axis [114]. The hypothalamus releases gonadotropin-releasing hormone (GnRH), which reaches the adenohypophysis through the hypophyseal portal system. As a response, the GnRH triggers the release of gonadotropins (luteinizing hormone (LH), and follicle-stimulating hormone (FSH)), which act at the testicular level. LH stimulates the Leydig cells to produce testosterone [114]. LH binding triggers the internalization of cholesterol, which is converted to pregnenolone, and then to 17α-hydroxy-pregnenolone, and androgen dehydroepiandrosterone (DHEA). DHEA is converted to androstenedione via 3β-hydroxysteroid dehydrogenase (3β-HSD), and then to testosterone via 17β-hydroxysteroid dehydrogenase-3 (17β-HSD-3) [115]. FSH binds on the Sertoli cells, supporting the spermatogenesis. This process includes three phases of development: (1) The proliferation of spermatogonial cellular precursors (mitosis); (2) Maturation into spermatocytes, with recombination, reduction, and division of DNA (meiosis); and (3) Differentiation into spermatids, and, finally, into mature spermatozoa (spermiogenesis) [116]. 

Alcohol consumption seems to affect endocrine functions, compromising the regulation of the HPG axis; however, contradictory evidence is reported in the literature. In 2005, a study investigating fertility hormones in 66 heavy drinkers reported increased levels of FSH, LH, and estradiol [79]. Conversely, Maneesh et al. reported lower levels of the gonadotropins, FSH and LH, in 45 alcoholic men [80], while Jensen et al. did not observe any association between the levels of FSH, LH, and inhibin B and the intake of alcohol in 8344 healthy men [81]. Similar results are reported regarding testosterone concentration, with studies observing higher values in alcoholic men [81,82,83], while others observed lower testosterone [79,80]. In addition, chronic alcohol intake can increase serum prolactin (hyperprolactinemia), causing hypogonadism, reduced sperm production, impotence, and gynecomastia in men [117,118]. These differences could be due to the heterogeneity in the study designs, the populations included, and the different approaches to classifying drinking habits, whereas the studies conducted in animal models reported a reduction in the circulating levels of LH and FSH following EtOH exposure [67,68,69,70,71,72]. 

Studies dating back to the 1980s suggest that acute EtOH intake might directly inhibit the hypothalamic release of GnRH, and, consequently, reduce the levels of LH and testosterone [119,120], probably through the alteration of the nitric oxide (NO) pathway. NO stimulates the synthesis of PGE_2_ in tissues by binding the heme group of cyclooxygenase-1 [121], while EtOH inhibits the cyclooxygenase-1 activity, and eventually the synthesis of PGE_2_ and GnRH secretion [122]. Alcohol consumption has also been associated with a reduction in the number of Leydig cells and an altered morphology [78]. The Leydig cells of rats fed with EtOH showed reduced sizes, swollen mitochondria, and smaller amounts of cytosol, along with the reduced synthesis of testosterone [78]. Besides altering its synthesis, alcohol consumption is also associated with an enhanced rate of testosterone elimination [17,73]. In fact, EtOH stimulates the activity of aromatase, which converts testosterone into estradiol [73], resulting in elevated estrogen levels and abnormal breast enlargement, as observed in heavy alcoholics [17,83]. This further inhibits the synthesis of FSH and LH, and, consequently, the synthesis of testosterone itself [74,75]. Moreover, chronic alcohol intake has been associated with the reduced bioavailability of insulin growth factor (IGF)-1, which physiologically stimulates the synthesis of testosterone [76]. Finally, it is important to mention that NAD is the cofactor of enzymes involved in both EtOH and androgen metabolism. Therefore, high EtOH intake reduces NAD^+^/NADH levels, and indirectly inhibits the activity of enzymes involved in testosterone synthesis [77]. 

### 4.2. Impact of Alcohol Consumption on Semen Quality

Data from an animal study showed that an EtOH-rich diet can affect testicular function, with consequences on the semen quality. In fact, EtOH-fed mice showed compromised integrities of the testis and seminal vesicles, and altered weight of the prostate, which resulted in increased germ cell desquamation, decreased sperm concentrations, and increased abnormal sperm morphologies [123]. Besides the alterations in the semen quality (lower sperm concentration, motility, and percentage of normal forms), Rahimipour et al. also reported reduced DNA condensation and integrity in mice fed with ethanol compared to controls, along with increased apoptotic rates [85]. In addition, in vitro experiments showed an accelerated acrosomal loss occurring during the sperm capacitation of human and animal sperm incubated in ethanol, further reducing their fertilizing ability [124,125,126,127]. This is probably due to the capacity of ethanol to alter lipid fluidity and membrane permeability through the oxidation of the membranes’ lipids and proteins [127]. In rats, decreased sperm motility was observed after exposure to EtOH, as well as changes in the meiotic divisions, reduced gametes viability, and a higher rate of sperm with poorly condensed chromatin [86,87]. In humans, a case study reported severe oligoasthenoteratozoospermia in an alcoholic man, which evolved into cryptozoospermia, and then azoospermia after a few years [91]. In 2017, a meta-analysis investigated the impact of alcohol intake on semen quality by analyzing evidence from 18 cross-sectional studies [93]. The authors concluded that daily alcohol consumption results in a worsened semen quality, particularly in terms of the semen volume and the sperm morphology. However, this effect was not reported for occasional drinkers, while the authors observed even better sperm motility in occasional drinkers than never drinkers, despite all the limitations identified in their analysis [93]. In fact, the association between semen quality and the amount of alcohol consumed is still controversial. Surprisingly, Ricci et al. observed a positive correlation between semen volume and concentration, and moderate alcohol consumption (equal to 4–7 units/week), suggesting that a limited consumption of alcohol may improve semen quality [96]. This might be explained by the fact that some compounds present in alcoholics drinks (i.e., natural flavonoids, and polyphenols in red wine) have antioxidant and anti-inflammatory activities, and they reportedly have a positive influence on semen quality (particularly by improving sperm motility, concentration, and survival) at low concentrations [128,129,130,131]. However, a cross-sectional study including 8344 healthy men did not report any association between low/moderate alcohol consumption and semen quality [81]. Similarly, other studies failed to identify any coherent dose–response pattern in the semen parameters depending on the degree of alcohol consumption [83,88,132,133,134]. Boeri et al. suggested that the correlation between alcohol consumption and alterations in the semen parameters might be directly proportional to the amount of alcohol consumed. In fact, the semen parameters were reportedly worse in samples of heavy rather than moderate drinkers [89]. Several recent studies of different global geographic regions have confirmed the negative impact of heavy alcohol consumption on semen quality. In fact, in China, a cross-sectional study conducted in 2020 reported reduced sperm concentrations in 55 heavy drinkers suffering from secondary infertility [88], while in Italy, 45 heavy drinkers with primary infertility showed reduced sperm concentrations and motilities compared to moderate drinkers or abstainers [89]. Similarly, an inverse association between sperm counts and alcohol consumption was observed in a Brazilian population of 167 infertile men [90], while a large study conducted on a Danish population of 1221 men showed a direct association between worsening semen quality and increasing alcohol intake [92]. Other studies have also confirmed a higher rate of sperm DNA fragmentation and chromatin decondensation in heavy drinkers [89,90,94,95].

The differences in the study designs, and the discrepancies in the published studies, make it challenging to draw any conclusions regarding the association between the amount of alcohol consumed and the semen quality. Hence, much research is still warranted in this regard.

### 4.3. Impact of Alcohol Consumption on Gene Transcription, Genetics, and Epigenetics Regulation

Male infertility associated with chronic alcohol consumption might also be due to a differential regulation of gene expression, followed by an altered metabolism of the specific proteins involved in sperm maturation [97,98,101,135]. In fact, it was shown that EtOH can lead to oxidative damage in the epididymis by altering the mRNA expression of β-defensin, which has antimicrobial properties and is involved in sperm function [97]. Alcohol exposure also altered post-transcriptional RNA modifications in murine sperm, thereby influencing the expression of small mitochondrial RNA species, the mitochondrial function of spermatozoa, and the reproductive ability [101]. In addition, EtOH compromised sperm viability by reducing the expression of proliferating cell nuclear antigen (PCNA) in germ cells and by promoting apoptosis [135]. 

Several genetic variants of enzymes involved in EtOH metabolism have been identified, associated with different degrees of tolerability to alcohol. In fact, specific single nucleotide polymorphisms in ADH and ALDH show altered enzymatic activity, which results in the accumulation of acetaldehyde. Because of its unpleasant effects, this is associated with a lower risk of developing an alcohol-use disorder [101].

A correlation between EtOH consumption and epigenetic changes in sperm DNA has been explored in various studies [100,101,102,103,136]. In this regard, aberrant gene methylation in sperm DNA was associated with male infertility [102]. During early embryonic development, there are two loci in paternal DNA (differentially methylated region upstream of the H19 gene-H19 DMR, and intergenic differentially methylated region, IG-DMR) that are highly methylated, and that are fundamental for growth and neurobehavioral development [103]. A significant difference was observed in the demethylation rates of specific C-phosphate-G (CpG) sites between nondrinkers and moderate drinkers. Particularly, there was a direct correlation between the amount of alcohol consumed and the degree of demethylation at the H19 DMR and IG-DMR genes, supporting the hypothesis that EtOH consumption may reduce the DNA methyltransferase activity, and increase the risk of the transmission of defective imprinted genes [103]. In addition, as shown in the rat testis, EtOH exposure may affect spermatogenesis, as it increases the acetylation of lysine 9 in histone 3 (Ac-H3-lys9), which results in impaired sperm chromatin organization and embryonic development [100]. 

### 4.4. Consequences of Paternal Alcohol Consumption on the Offspring

Several studies were conducted in animal models to investigate the transgenerational effect of paternal alcohol exposure, showing low fetal and birth weights in the offspring, as well as altered organ weights and hormonal regulations (for a review on the topic, see [101]). An animal study showed that paternal alcohol exposure led to hormonal and nervous system anomalies in the offspring. Specifically, the expression of nerve growth factor (NGF), a well-characterized neurotrophin involved in the development of the nervous system in vertebrates [105], was strongly reduced in the frontal cortices of the offspring of EtOH-fed mice, as well as in the hippocampal, hypothalamic, and olfactory lobes, leading to the conclusion that paternal alcohol consumption can produce critical alterations in the brains of offspring [106]. The anogenital distance, which is a male fertility marker related to the proper functioning of the endocrine system, was also shorter in the offspring of EtOH-consuming fathers, indicating that alcohol may have an adverse effect on the reproductive development of offspring [107]. Paternal alcoholism before conception was also associated with limited offspring growth and decreased placental efficiency [104].

In a case of paternal alcohol exposure, the offspring showed reproductive dysfunctions similar to those reported for direct alcohol consumption, including alterations in the hormonal axis and the semen quality [113]. 

Three possible mechanisms have been proposed to explain the effect of paternal alcohol consumption on the offspring: (a) An alteration of the sperm chemical composition, leading to behavioral, biochemical, and hormonal disturbances in the offspring; (b) A failure of the elimination of EtOH-damaged sperm; and (c) An induction of genetic mutations in sperm DNA that can be transmitted to the offspring [137]. Although chemical alterations in sperm and/or seminal plasma may compromise embryonic development, this remains to be confirmed. The second hypothesis refers to the mechanisms of natural selection that may fail to remove spermatozoa damaged by EtOH during spermatogenesis, in favor of genetically intact spermatozoa. The third hypothesis emphasizes a heritable genetic change in EtOH consumers, and seems to be the most accepted [137]. However, a study published in 2017 was not able to identify any association between the phenotypes observed in mice exposed to EtOH (fetal growth restriction and altered developmental programming) and the paternal DNA methylation profiles, questioning the transgenerational effect of EtOH exposure [138]. EtOH-related epigenetic effects on the paternal germline might provide an explanation for the transgenerational influence of the father’s lifestyle habits on the development of the offspring, and it surely deserves more investigation [139].

## 5. Limitations of the Published Human Studies and Future Areas of Investigation

Studies on the effects of alcohol dependence in the human population suffer from significant limitations. This is due mostly to the fact that the data is self-reported and collected by using questionnaires [140]. In fact, the objective estimation of the real volume of alcohol consumed is challenging because of the possibility of an over- or underestimation [141]. Moreover, alcoholics often do not admit to having an alcohol dependence, hence leading to incorrect information about the amount of alcohol and the frequency of the drinking. The investigation is made even more challenging by the fact that alcohol consumption can be defined according to different criteria (such as frequency, or units/daily), and by the fact that the pattern of alcohol consumption can vary, from moderate/heavy alcohol consumption to binge drinking. This can influence the analyzed outcomes and can make it difficult to compare results from different publications [142]. 

The negative impact of EtOH on male fertility is a confirmed hypothesis; however, the molecular mechanisms that act at the sperm level remain to be clarified. In fact, studies analyzing the impact of alcohol consumption on male infertility are difficult to plan because semen quality is influenced by a multitude of other factors and varies significantly between individuals, and in the same individual as well [143]. Hence, there is a need to identify molecular markers for alcohol-related male infertility. In this context, molecular tests to assess advanced sperm characteristics (i.e., acrosome reaction, capacitation) are to be considered valid approaches for discovering biomarkers, besides the conventional semen parameters. Similarly, -OMICS sciences are to be kept in consideration. Comparative proteomic studies showed the differential expression of proteins between normozoospermic and abnormal semen samples [144], and were able to successfully identify several proteins in human sperm and seminal fluid as possible markers of male infertility and reproductive failure [144,145]. In asthenozoospermia, the altered expression of cAMP-mediated protein kinase A and actin was reported, which would explain the low motility of spermatozoa. Similarly, a proteomic screening of globozoospermic cells revealed reduced levels of α-tubulin, β-tubulin, and β-galactoside–galactoside binding protein and vimentin, indicating a weakened cytoskeletal organization, which may contribute to the pathogenesis of globozoospermia [144,146]. Furthermore, proteomic analyses of seminal plasma have allowed for the identification of the markers for obstructive and nonobstructive azoospermia (as reviewed in [147]). Hence, the high-throughput analysis of semen samples is extremely useful, both in diagnostics and for identifying new therapeutic targets, and may be successfully applied to the investigation of alcohol-induced male infertility. In association with proteomic studies, transcriptomic studies can help to identify the molecular markers of male infertility. The transcriptome is defined as the overall RNA content of a sample, including the microRNA, whereas the alteration of several microRNAs has been associated with EtOH-related damage and tolerance [148], and their expression in the human semen samples of alcoholics has not yet been investigated. On the other hand, in mice, sperm RNA sequencing showed the altered enrichment of microRNAs after EtOH exposure, suggesting that they play a role in the alcohol-related phenotypes of the offspring [149]. Furthermore, different sites can generate extracellular microvesicles (exosomes), including RNAs and proteins, which are endocytosed by sperm [150,151]. The molecular composition of exosomes is a picture of the status of their cellular origin; hence, their characterization may be helpful to identifying novel site-specific pathological biomarkers [151]. Similarly, a metabolomics analysis of seminal plasma has allowed for the identification of the metabolites involved in energy production, the maintenance of the redox status, hormonal regulation, and sperm functions [144], representing a promising approach for investigating EtOH-related male infertility. 

The ability to conduct genome-wide association studies has allowed for the scanning of the entire genome for the identification of any novel genetic locus associated with specific clinical conditions. In this context, the creation of consortia has allowed for the analysis of a large population with alcohol dependence [152]. A study conducted in 2018 analyzed around 15,000 subjects with alcohol dependence, and identified functional variants of the ADH1B gene that were differentially presented in European and African populations, and that were associated with a variable rate of EtOH oxidation, and protection from EtOH-related side effects and the risk for developing alcohol dependence [153]. Similarly, a recent study investigating more than 270,000 multiethnic subjects identified different genetic loci associated with heavy drinking behavior or the risk of developing alcohol dependence [154]. Besides the genes involved in EtOH metabolism, these studies have also identified new genes (i.e., β-klotho (KLB), and fibroblast growth factor 21 (FGF21)), whose alleles are strongly associated with alcohol consumption [153,154]. 

In the future, the use of an integrated OMICS approach could explain the mechanisms underlying EtOH-induced infertility due to alterations in the semen parameters and the redox status, as well as disturbances in hormonal regulation, and could allow for the identification of selective markers in these subjects. Furthermore, considering the role of oxidative stress in mediating EtOH-related damage, further human studies should be planned to better understand the mechanisms involved, and to investigate the possible treatment with an adequate therapy.

## 6. Conclusions

Besides being an important public and social issue, alcohol consumption can also significantly impact male reproduction. The association between chronic alcohol consumption and poor semen quality has been reported in a large number of studies in both humans and animals, mainly due to excessive ROS generation following EtOH metabolism. By acting as genotoxic agents, EtOH and its metabolites alter the expression of specific genes involved in the hormonal regulation of spermatogenesis, and increase sperm DNA fragmentation, potentially with a transgenerational effect on the offspring. Despite its role in contributing to male infertility, specific markers for EtOH-related infertility have not yet been identified. In this context, high-throughput analyses of semen samples will represent the future strategies for a better understanding of the molecular mechanisms underlying alcohol-induced male reproductive disorders, as well as for identifying more effective therapeutic measures to combat the dependence.

## Figures and Tables

**Figure 1 ijerph-19-00328-f001:**
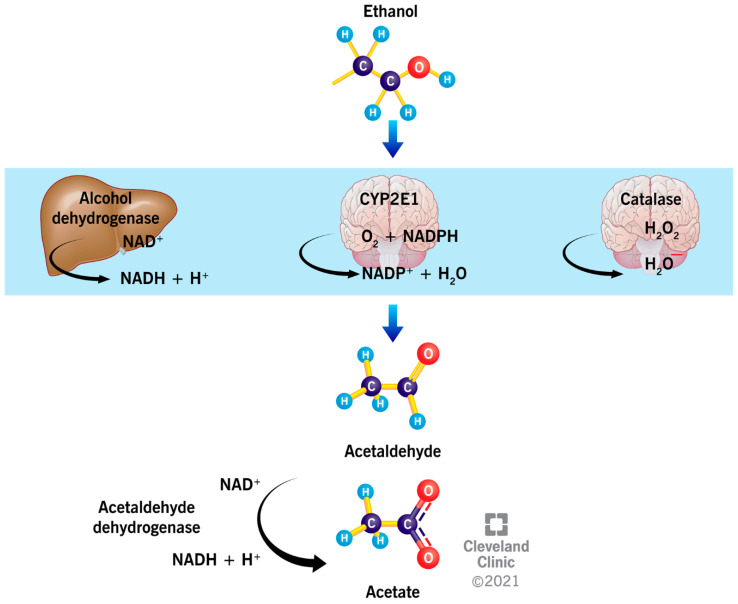
The metabolism of ethanol. In the liver, ethanol is oxidized to acetaldehyde mainly by the enzyme, alcohol dehydrogenase, while, in the brain, the enzymatic activity of cytochrome P450 2E1 (CYP2E1) and catalase are more prominent. Acetaldehyde is further oxidized to acetate by the enzyme, acetaldehyde dehydrogenase.

**Figure 2 ijerph-19-00328-f002:**
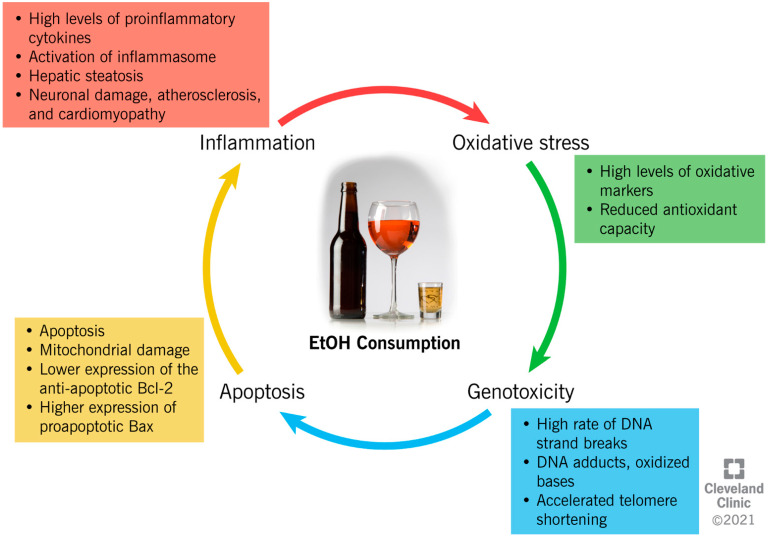
Ethanol-induced mechanisms of cellular damage.

**Table 1 ijerph-19-00328-t001:** Summary of the evidence regarding the impact of alcohol consumption on male fertility in animal and human studies.

Impact of Alcohol	Animal Studies	Human Studies
**Effects on Reproductive Hormonal Regulation**	Reduced levels of LH, FSH [67,68,69,70,71,72]. Reduced levels of testosterone [73,74,75,76,77]. Altered Leydig cell number and morphology [78].	Contradictory evidence in literature on levels of FSH, LH, and testosterone [79,80,81,82,83].
**Effects on Semen Quality**	Reduced sperm concentration and motility [84,85,86,87]. Increased abnormal sperm morphology [84,85,86,87]. Defects in chromatin condensation [86,87].	Reduced sperm concentration [88,89,90]. Altered semen volume and increased abnormal sperm morphology [91,92,93]. Increased sperm DNA fragmentation and defects in chromatin condensation [89,90,94,95]. Moderate consumption associated with better semen volume and concentration [96].
**Effects on Gene Transcription, Genetic, and Epigenetic Regulation**	Altered expression of RNA involved in sperm function [97,98]. Altered expression of proteins involved in apoptosis [99]. Aberrant gene acetylation of sperm DNA [100].	Altered expression of RNA involved in sperm function [101]. Aberrant gene methylation in sperm DNA [102,103].
**Transgenerational Effects**	Low fetal and birth weight, and limited growth in offspring [101,104]. Nervous system anomalies in offspring [105,106]. Altered reproductive development of offspring [107].	Higher incidence of psychopathological disorders [108,109,110], congenital heart defects [111], cancer [112], and altered reproductive development [113] in the offspring.

## Data Availability

Not applicable.

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
