# Peer review of "Impact of Alcohol Consumption on Male Fertility Potential: A Narrative Review"

_ijerph, 2021, doi:10.3390/ijerph19010328_

Reviewer 2 Report

Well conducted review

Worth publishing

This is an extensive review of the current literature regarding alcohol consumption and its relation to male fertility.
The review is comprehensive including studies in animals and humans. 
Pathophysiology and basic research are negotiated.
The lack of RCTs and the impact to live birth rate is highlighted
Although this review does not add to the current evidence, it is well wtitten and worth of publishing.

Author Response

Reviewer 2

Well conducted review

Worth publishing

This is an extensive review of the current literature regarding alcohol consumption and its relation to male fertility.
The review is comprehensive including studies in animals and humans. 
Pathophysiology and basic research are negotiated.
The lack of RCTs and the impact to live birth rate is highlighted
Although this review does not add to the current evidence, it is well written and worth of publishing.

A: Authors thank Reviewer 2 for his/her positive feedback.

Reviewer 3 Report

Dear authors,

Thank you for the paper submitted covering an interesting topic. I think this article tries to examine the effect of alcohol consumption on male fertility, which is an interesting topic.

The manuscript should be strongly improved, as I detail in the following specific areas:

Title: The type of the review is not identified in the title.

Abstract:

Neither the main results nor the conclusions are described in this section. I suggest being more specific about this.

Introduction:

Lines 36-42: This paragraph is a falsehood and should be removed. There is no safe alcohol consumption level, so total avoidance is the best option. It is not possible to preconize that alcohol consumption is beneficial, reducing the risk of several diseases. The references that endorse these affirmations are out-dated: The most recent one is from 2005. At that time, the alcohol lobby supported alcohol consumption. Nowadays, the fact of moderate drinking as a preventive health strategy cannot be published in any health-related journal.

Figure 2: The font size should be enlarged.

The need for the current review is not well explained.

The harmful effects that alcohol consumption produces in the body are extensively developed, moving away from the main topic. And it is not until Section 4 that this topic is discussed  

After the introduction, there is no Methods section in order to know the eligible criteria (year of dissemination, language…). Also, there are no specifications for the ineligible criteria, publication status, study design, and databases of coverage). In this rather unsystematic approach, selection of information from primary articles is usually subjective, lacks explicit criteria for inclusion and leads to a biased review. When this occurs it is difficult to discern if the author has constructed an objective review of the literature or a lengthy commentary.

There is no reference to the information sources, search strategy, selection process, data collection process, etc.

The results section is mandatory, to know the flow of studies, the study characteristics,...

There is no discussion section. There is no general interpretation of the results in the context of evidence.

The references are not updated, and almost half of them are older than five years. In fact some of them are from the 1970´s.

Author Response

Reviewer 3

Dear authors,

Thank you for the paper submitted covering an interesting topic. I think this article tries to examine the effect of alcohol consumption on male fertility, which is an interesting topic.

The manuscript should be strongly improved, as I detail in the following specific areas:

Title: The type of the review is not identified in the title.

A: The title has been revised as follows: Investigating the Impact of Alcohol Consumption on Male Fertility Potential – A Narrative Review.

Abstract:

Neither the main results nor the conclusions are described in this section. I suggest being more specific about this.

A: We improved the abstract by including main results and conclusions (page 1, lines 22-26).

Introduction:

Lines 36-42: This paragraph is a falsehood and should be removed. There is no safe alcohol consumption level, so total avoidance is the best option. It is not possible to preconize that alcohol consumption is beneficial, reducing the risk of several diseases. The references that endorse these affirmations are out-dated: The most recent one is from 2005. At that time, the alcohol lobby supported alcohol consumption. Nowadays, the fact of moderate drinking as a preventive health strategy cannot be published in any health-related journal.

A: Lines 40-43 (page 1) have been deleted based on Reviewer’s feedback to remove any reference to safe alcohol consumption level. However, we would like to highlight that few studies have been recently published showing the beneficial effects on semen of alcohol consumption and exposure to compounds present in wine in modest quantities:

  1. Aquila et al., Red Wine Consumption May Affect Sperm Biology, Molecular Reproduction & Development, 2013.

Original article: The authors studied estrogen-like effect of the phytoestrogen myricetin (natural flavonoid, particularly enriched in red wines) on human ejaculated sperm biology. They reported 1) Exposure to low concentrations of Myricetin is able to improve sperm motility and survival, whereas higher concentrations tend to be less effective. 2) Myricetin triggers capacitation and acrosome reaction.

  1. Ricci et al., Alcohol intake and semen variables: cross-sectional analysis of a prospective cohort study of men referring to an Italian Fertility Clinic, Andrology 2018

Original article: The authors investigate the relation between life-style and fertility. 1) U shaped association between alcohol consumption and sperm concentration was confirmed, after adjusting to potential cofounders; 2) Higher semen volume was observed in men with 4–7 units/week of alcohol intake, and ≥8 units/week were not negatively associated with other seminal variables.

  1. Ricci et al., Semen quality and alcohol intake: a systematic review and meta-analysis. Reproductive BioMedicine Online, 2017

Systemic review and Meta-analysis: The authors investigated the association between alcohol intake and semen quality. They reported that 1) When comparing occasional versus never drinkers, alcohol was shown to have a statistically significant positive effect on sperm motility; 2) Morphology was better in occasional drinkers compared with daily drinkers.

  1. Mongioi et al., The Role of Resveratrol in Human Male Fertility, Molecules, 2021

Review Article: The authors evaluated the effect of RSV (natural non-flavonoid polyphenol widely present in the Mediterranean diet including wine) on human male fertility and the mechanisms by which it could act on human spermatozoa. RSV at low concentrations has a positive effect on sperm motility, whereas at higher concentrations it has a detrimental effect on this parameter. RSV has a positive effect on sperm concentration, total sperm count, and both total and progressive motility, without affecting sperm morphology, ejaculate volume, and PH.

Figure 2: The font size should be enlarged.

A: Both figures have been improved by the Arts department of Cleveland Clinic, and the font size increased.

The need for the current review is not well explained.

A: It is important to understand the mechanisms leading to alcohol-related male infertility as a large percentage of male alcoholics are in the reproductive age and may be interested in achieving a pregnancy. This has been clarified at page 2, lines 64-67.

The harmful effects that alcohol consumption produces in the body are extensively developed, moving away from the main topic. And it is not until Section 4 that this topic is discussed  

A: Based on Reviewer’s feedback, we have significantly shortened the introductory sections 1-3. In addition, we have created a single shorter Section 3, by combining all the subsections.

After the introduction, there is no Methods section in order to know the eligible criteria (year of dissemination, language…). Also, there are no specifications for the ineligible criteria, publication status, study design, and databases of coverage). In this rather unsystematic approach, selection of information from primary articles is usually subjective, lacks explicit criteria for inclusion and leads to a biased review. When this occurs it is difficult to discern if the author has constructed an objective review of the literature or a lengthy commentary.

There is no reference to the information sources, search strategy, selection process, data collection process, etc.

The results section is mandatory, to know the flow of studies, the study characteristics,...

There is no discussion section. There is no general interpretation of the results in the context of evidence.

A: We agree with the Reviewer that the major advantage of systematic reviews is that they are based on the findings of comprehensive and systematic literature searches in all available resources, with minimization of subjective selection bias. On the other side, narrative reviews, if they are written by experts in certain research area, can provide experts' intuitive, experiential, and explicit perspectives in focused topics. Moreover, they are useful educational articles since they pull many pieces of information together into a readable format. While systematic reviews are superior to narrative reviews in answering specific questions, narrative reviews are better suited to addressing a topic in wider ways and providing more potential for individual insight and opportunities for speculation than most quantitative review approaches.

Our study wants to be an unsystematic narrative review written by international experts in male infertility and cytotoxicity. This explains why we did not structure the article as a systematic review including methods - results - discussion sections, and we did not discuss the characteristics of any single study individually. The aim of this review is to present a broad perspective on the topic of alcohol-associated male infertility in a narrative way.

The references are not updated, and almost half of them are older than five years. In fact some of them are from the 1970´s.

A: We tried to discuss published evidence on the topic by including animal and human studies, although many studies were older than 5 years. Overall, this study includes 149 references, from 1970s (n=7), 1980s (n=7), 1990s (n=10), 2000-2009 (n=24), 2010-2019 (n=87), 2020-2021 (n=14).

Reviewer 4 Report

 Investigating the Impact of Alcohol Consumption on Male Fertility Potential  

       Abstract: Alcohol abuse disorder is a serious condition, implicating more than 15 million people aged and older in 2019 in the United States. Ethanol (or ethyl alcohol) is mainly oxidized in the liver, resulting in the synthesis of acetaldehyde and acetate, toxic and carcinogenic metabolites, and the generation of a reductive cellular environment. Moreover, ethanol can interact with lipids, generating afatty acid ethyl esters and phosphatidylethanol, which interfere with physiological cellular pathways. This narrative review summarizes the impact of excessive alcohol consumption on male fertility by describing its metabolism and how ethanol consumption may induce cellular damage. Further, the impact of alcohol consumption on hormonal regulation, semen quality, and genetic and epigenetic regulations, is discussed based on evidence from animal and human studies, with a focus on the consequences for the offspring. Finally, limitations of the current evidence are discussed, while new landscapes of investigation are proposed for the identification of molecular markers for alcohol-associated infertility, with a focus on advanced -OMICS based approaches applied for the 21 analysis of semen samples.

-----------------------------

The authors presented an overall review of the potential impact of alcohol consumption of male fertility – gross, cellular and molecular/genetic.

Comments

    Topic:

    The topic is misleading, though inclusive of entire content of the manuscript. This is a review paper.

         Suggestion: “Impact of Alcohol Consumption on Male Fertility Potential. Review 

“Impact of alcohol on sex hormonal regulation”

  1. Brief description of the process of spermatogenesis and steroidogenesis
  2. The HPG- axis and GnHRH, LH, FSH, testosterone etc (HPG = hypothalamic-Pituitary-Gonadal)

Morphological characteristics alone cannot assess the degree of sperm fertilizing capabilities. The assessment of alcohol on acrosomal reaction/capacitation is recommended/needed. Some figures could augment the interest of readers. Furthermore, the outcome could be an addition to the search for new molecular markers  

The conclusions are scientifically sound and consistent with the data presented

---------------------------------------------------------------------------------

Author Response

Reviewer 4

Investigating the Impact of Alcohol Consumption on Male Fertility Potential  

       Abstract: Alcohol abuse disorder is a serious condition, implicating more than 15 million people aged and older in 2019 in the United States. Ethanol (or ethyl alcohol) is mainly oxidized in the liver, resulting in the synthesis of acetaldehyde and acetate, toxic and carcinogenic metabolites, and the generation of a reductive cellular environment. Moreover, ethanol can interact with lipids, generating afatty acid ethyl esters and phosphatidylethanol, which interfere with physiological cellular pathways. This narrative review summarizes the impact of excessive alcohol consumption on male fertility by describing its metabolism and how ethanol consumption may induce cellular damage. Further, the impact of alcohol consumption on hormonal regulation, semen quality, and genetic and epigenetic regulations, is discussed based on evidence from animal and human studies, with a focus on the consequences for the offspring. Finally, limitations of the current evidence are discussed, while new landscapes of investigation are proposed for the identification of molecular markers for alcohol-associated infertility, with a focus on advanced -OMICS based approaches applied for the 21 analysis of semen samples.

-----------------------------

The authors presented an overall review of the potential impact of alcohol consumption of male fertility – gross, cellular and molecular/genetic.

 A: Authors thank Reviewer 4 for the positive feedback.

Comments

     Topic:

    The topic is misleading, though inclusive of entire content of the manuscript. This is a review paper.

         Suggestion: “Impact of Alcohol Consumption on Male Fertility Potential. Review”  

A: The title has been revised as follows: Investigating the Impact of Alcohol Consumption on Male Fertility Potential – A Narrative Review.

“Impact of alcohol on sex hormonal regulation”

  1. Brief description of the process of spermatogenesis and steroidogenesis
  2. The HPG- axis and GnHRH, LH, FSH, testosterone etc (HPG = hypothalamic-Pituitary-Gonadal)

A: We have included a brief description of the HPG axis, the process of spermatogenesis and steroidogenesis (page 9, lines 281-295).

Morphological characteristics alone cannot assess the degree of sperm fertilizing capabilities. The assessment of alcohol on acrosomal reaction/capacitation is recommended/needed. Some figures could augment the interest of readers. Furthermore, the outcome could be an addition to the search for new molecular markers  

A: Authors thank Reviewer 4 for this suggestion. We have discussed the impact of ethanol on this outcome at page 10, lines 337-341, in the section 4.2. Impact of alcohol consumption on semen quality. The studies identified on this topic are limited, dated, and mostly conducted in animal models, and they do not add much to the discussion on alcohol-related male infertility. For this reason, we prefer not to include an additional figure.

The investigation of this outcome for the identification of new molecular markers has been proposed (page 12, lines 462-465).

The conclusions are scientifically sound and consistent with the data presented

 A: Authors thank Reviewer 4 for his/her positive feedback.

Round 2

Reviewer 3 Report

Dear authors, 

This article does not follow the recommendations that a narrative review should follow to be published. Although the authors have insisted that the paper has the structure of an unsystematic narrative review written by international experts in male infertility and cytotoxicity, I think it is mandatory to follow the rules that a JCR peer review journal requieres.  I still think that this structure fits more to a chapter of a book than to a scientific article so I encourage the authors to publish a book about this topic.